# Plasmonic nano-aperture label-free imaging of single small extracellular vesicles for cancer detection
Nareg Ohannesian[1,7], Mohammad Sadman Mallick[1,7], Jianzhong He[2], Yawei Qiao[2], Nan Li[2], Simona F. Shaitelman[2], Chad Tang[2], Eileen H. Shinn [3], Wayne L. Hofstetter[4], Alexei Goltsov[4], Manal M. Hassan[5], Kelly K. Hunt [6], Steven H. Lin [2]✉ & Wei-Chuan Shih[1]✉

## Abstract

**Background** Small extracellular vesicle (sEV) analysis can potentially improve cancer detection and diagnostics. However, this potential has been constrained by insufficient sensitivity, dynamic range, and the need for complex labeling.
**Methods** In this study, we demonstrate the combination of PANORAMA and fluorescence imaging for single sEV analysis. The co-acquisition of PANORAMA and fluorescence images enables label-free visualization, enumeration, size determination, and enables detection of cargo microRNAs (*miRs*).
**Results** An increased sEV count is observed in human plasma samples from patients with cancer, regardless of cancer type. The cargo *miR-21* provides molecular specificity within the same sEV population at the single unit level, which pinpoints the sEVs subset of cancer origin. Using cancer cells-implanted animals, cancer-specific sEVs from 20 μl of plasma can be detected before tumors were palpable. The level plateaus between 5–15 absolute sEV count (ASC) per μl with tumors ≥8 mm$^3$. In healthy human individuals ($N = 106$), the levels are on average 1.5 ASC/μl ($+/-$ 0.95) without *miR-21* expression. However, for stage I–III cancer patients ($N = 205$), nearly all (204 out of 205) have levels exceeding 3.5 ASC/μl with an average of 12.2 ASC/μl (±9.6), and a variable proportion of *miR-21* labeling among different tumor types with 100% cancer specificity. Using a threshold of 3.5 ASC/μl to test a separate sample set in a blinded fashion yields accurate classification of healthy individuals from cancer patients.
**Conclusions** Our techniques and findings can impact the understanding of cancer biology and the development of new cancer detection and diagnostic technologies.

## Plain Language Summary

Small extracellular vesicles (sEVs) are tiny particles derived from cells that can be detected in bodily fluids such as blood. Detecting sEVs and analyzing their contents may potentially help us to diagnose disease, for example by observing differences in sEV numbers or contents in the blood of patients with cancer versus healthy people. Here, we combine two imaging methods – our previously developed method PANORAMA and imaging of fluorescence emitted by sEVs —to visualize and count sEVs, determine their size, and analyze their cargo. We observe differences in sEV numbers and cargo in samples taken from healthy people versus people with cancer and are able to differentiate these two populations based on our analysis of sEVs. With further testing, our approach may be a useful tool for cancer diagnosis and provide insights into the biology of cancer and sEVs.

A prominent factor that influences cancer prognosis is early detection. While current screening tests used for several prevalent cancers improve outcomes by enhancing early detection of cancers such as lung, breast, and colon cancers, these tests suffer in sensitivity or specificity, leading to either inadequate or inappropriate diagnostic evaluations, respectively. Non-invasive testing through liquid biopsy, which examines tumor-derived biomarkers in physiological fluids such as blood and saliva, may improve diagnostic testing by detecting shed tumor components, such as circulating tumor DNA (CTDNA), circulating tumor cells (CTC), and circulating tumor sEVs[1–4]. While blood tests that quantify and characterize CTDNA

[1]Department of Electrical and Computer Engineering, University of Houston, 4800 Martin Luther King Blvd., Houston, TX 77204, USA. [2]Department of Radiation Oncology, The University of Texas MD Anderson Cancer Center, 1515 Holcombe Blvd, Houston, TX 77030, USA. [3]Department of Behavioral Science, The University of Texas MD Anderson Cancer Center, 1515 Holcombe Blvd, Houston, TX 77030, USA. [4]Department of Thoracic and Cardiovascular Surgery, The University of Texas MD Anderson Cancer Center, 1515 Holcombe Blvd., Houston, TX 77030, USA. [5]Department of Epidemiology, The University of Texas MD Anderson Cancer Center, 1515 Holcombe Blvd., Houston, TX 77030, USA. [6]Department of Breast Surgical Oncology, The University of Texas MD Anderson Cancer Center, 1515 Holcombe Blvd., Houston, TX 77030, USA. [7]These authors contributed equally: Nareg Ohannesian, Mohammad Sadman Mallick.
✉e-mail: shlin@mdanderson.org; wshih@central.uh.edu

and CTC in circulation are highly specific and are important tools to help guide drug therapy by identifying targetable tumor mutations, early-stage cancers do not shed DNA or cells in sufficient levels to be easily detectable[5,6]. In contrast, sEVs, which include exosomes, microvesicles and more, are actively excreted by viable mammalian cells, making them more common than shed cells or DNA, especially in the earliest stages[3,7,8]. Exosomes are sEVs ranging from 30 to 200 nm responsible for intercellular communication. Exosomes also contain proteins, lipids, and nucleic acids that reflect the parental tumor cell[2,9,10], making it a valuable biomarker for cancer diagnosis[11]. However, owing to their nanoscale size, exosomes are generally "invisible" to most analytical tools due to insufficient scattered light to be detected[1,7]. To date, single sEV detection, characterization, and molecular profiling remain a challenge.

Our group has pioneered a nanostructured plasmonic metasurface by exploring the synergy of radiative coupling among individual gold nanodisks and precise removal of the substrate material underneath the gold nanodisks, called arrayed gold nanodisk on invisible substrate (AGNIS)[12], which exhibits improved sensitivity to local refractive index changes and enlarged sensing area. Using AGNIS, we have further developed plasmonic nano-aperture label-free imaging (PANORAMA) and demonstrated its capability of detecting individual nanoparticles as small as 25 nm[13]. Single sEV localization has been demonstrated recently using dynamic PANORAMA[14]. In this report, we demonstrate the application of PANORAMA to analyze individual antibody-captured sEVs for digital counting and sizing. We further demonstrate the integration of PANORAMA with fluorescence imaging to detect specific microRNAs (miR) contained within each sEV. Due to the co-acquisition of PANORAMA and fluorescence images, we can interrogate the molecular contents of each sEV already detected by PANORAMA. This capability enables us to pinpoint the sEV sub-population that are of cancer origin. Using human lung cancer xenograft models which secrete sEVs that express high levels of *miR-21* or that are green fluorescent protein (GFP) labeled, we found cancer-specific sEVs to be detectable in 20 μl plasma even before tumors became palpable and would reach steady state levels when tumors reach and exceed 8 mm³ (±4.25). This suggests that sEV concentrations may achieve different steady-state levels in normal vs. cancerous states and that assessing sEV concentration in the plasma could detect the presence of cancer. We were able to observe two distinct steady-state levels in healthy volunteers and patients with cancers of various origins and disease stages that could be used to accurately differentiate cancer patients from healthy individuals. Coupling measurement of absolute sEV concentration with molecular characterization could potentially achieve the level of sensitivity and specificity to enhance early-stage cancer detection.

## Methods
### Materials
PKH67 membrane labeling green dye was purchased from Millipore Sigma. SH-PEG-Biotin (1 KDa) and SH-PEG-CH₃ were purchased from Nanocs. Neutravidin (60 KDa) and biotinylated anti-human CD63; CD9; and CD81 antibodies were purchased from Thermo-Fisher Scientific and BioLegend, respectively. Ethanol (200 proof) was purchased from Decon Laboratories, Inc. Gold sputtering target was purchased from ACI Alloys, Inc. Argon gas (99.999%) was used for RF-sputter etching.

### Fabrication of AGNIS
Fabrication steps involve the deposition of Titanium (2 nm) as an adhesion layer and then the gold film (80 nm) using electron-beam evaporation. A monolayer of polystyrene beads of an average diameter of 460 nm (Thermo-Fisher Scientific) was assembled over the gold film. The substrate was exposed to oxygen plasma etching to shrink the size of the polystyrene beads, followed by Argon ion milling to etch away the uncovered part of the gold. The polystyrene beads were removed via sonication. This generated a two-dimensional polycrystalline array of gold nanodisks with an average diameter of 360 nm and a pitch (center-to-center distance) of 460 nm. This gold nanodisk array was immersed in a buffered HF (Sigma-Aldrich) to

undercut the glass substrate beneath the disks. The LSPR extinction peak of AGNIS is at 620 nm in air and 690 nm in water. The gold nanodisks occupy ~55% of the total substrate footprint. However, this is a conservative estimate because the LSPR electromagnetic field extends further away from the edges of the nanodisk. The 1/e distance for lateral electric field decay is ~30 nm, which brings the effective lateral imaging fill factor to ~75%[13].

### Surface functionalization
The AGNIS surface was first incubated with a mixture of polyethylene glycol (PEG) containing active Biotinylated Thiol-PEG (1 mM in PBS 1X) and inactive methylated Thiol-PEG (1 mM in PBS 1X) with a 1:3 mixture for 18 h. The surface was washed with DI water and next treated in 3.3 μM neutravidin solution to bind the neutravidin to biotin for 2 h, followed by a DI water rinse. The chip was then incubated in 50 μg/mL biotinylated antibody with 2.5% Blocking Albumin Serum (BSA) for 2 h, followed by DI water rinse.

### Small extracellular vesicles preparation and labeling
sEVs were collected from 5 mL of media from immortalized lung cancer cell culture (2–4 × 10⁶ cell/ml, cell line H460, ATCC), non-cancerous immortalized embryonal kidney cells (2–4 × 10⁶ cell/ml, cell line 293 A, ATCC), non-cancerous immortalized mammary epithelial cells (2–4 × 10⁶ cell/ml, cell line MCF10A, ATCC), non-cancerous bronchial epithelial (2–4 × 10⁶ cell/ml, primary cell line NHBE, ATCC). The culture media were collected, subjected to centrifugation at $800 \times g$ for 10 min to sediment the cells, and centrifuged at $12,000 \times g$ for 45 min to remove cellular debris. The sEVs were separated from the supernatant via centrifugation at $100,000 \times g$ for 2 h[15]. The sEVs pellet was washed once in a large volume of PBS and resuspended in 100 μL of PBS to yield the sEVs fraction. Ultracentrifugation does not provide the highest yield of sEVs, but it is one of the few techniques that provide the purest extraction with the most number of exosomes and least number of microvesicles and high-density protein aggregates[16–18]. The size distribution and concentration of extracted sEVs were quantified using NanoSight particle tracking (Malvern). The purified sEVs showed a size distribution ranging from 75 nm to 275 nm with a majority of particles at 115 nm (Supplementary Fig. S8). A portion of extracted sEVs was labeled with PKH67 following the protocol provided by the manufacturer. The sEV concentration used is 2 × 10⁵ sEV/μl unless stated otherwise.

### sEV isolation from cell culture medium for qPCR
H460 and 293 A cells were cultured in RPMI1640/DMEM with 10% FBS, then the serum-containing cell culture medium was removed, and the cells were cultured in an exosome free culture medium for 48 h. Then, the cell-culture supernatants were centrifuged at $2000 \times g$ for 10 min to remove debris, and large vesicles and apoptotic bodies were removed by centrifugation at $10,000 \times g$ for 30 min. Next, the sEVs were enriched by centrifugation at $3000 \times g$ for 30 min with the Amicon® Ultra Centrifugal Filter (Sigma catalog UFC9100) and then purified using Total Exosome Isolation Reagent (Thermo Fisher catalog 4478359) according to the manufacturer's recommendation. Isolated sEVs were resuspended with cold PBS and analyzed using a NanoSight NS300 (Malvern).

### RNA isolation from sEVs and qPCR
Purified sEVs from H460 and 293 A cells were subjected to Total Exosome RNA & Protein Isolation Kit (Thermo Fisher catalog 4478545) to extract total RNA. Followed by RT-PCR using OneTaq® RT-PCR Kit (New England Biolabs catalog E5310S). qPCR was done with the PowerUp™ SYBR™ Green Master Mix (Thermo Fisher catalog A25741). The thermal cycling protocol was as follows: initial denaturation for 10 min at 95 °C, followed by 50 cycles of 15 s at 95 °C and 60 s at 60 °C. The primer sequences are listed as follows:

*miR-21* for RT: 5′-CTCAACTGGTGTCGTGGAGTCGGCAATT-CAGTTGAGTCAACATC-3′

*miR-21* for qPCR: 5′-ACACTCCAGCTGGGTAGCTTATCA-GACTGA-3′

## Cell preparation and injection into mice

A549 and H460 frozen cells (one vial, conc. $1 \times 10^6$ cells/vial) were thawed in a T75 flask containing RPMI media with 10% FBS. Passage cells three times when cells reach 70–80% confluence. Cells were collected with Trypsin (TrypLE Express, gibco, 12605) and washed with PBS two times. Cancer cells were injected (0.1 ×10^6/100 μl to 1) into the right leg of the nude female mice (0002019 NU/J, Jackson Laboratory), 8-week-old at time of tumor implant, weighing around 25 grams each. This is an inbred strain with a spontaneous nonsense genetic mutation in the Foxn1 gene (forkhead box N1) on chromosome 11, from a single base pair (G) deletion in exon 3 which introduces a frameshift and a premature stop codon. Homozygous mice results in abnormal hair growth and defective development of the thymic epithelium. Homozygous nude mice lack T cells and suffer from lack of cell-mediated immunity, as well as B-cell development.

## Mouse tumor measurement

Mouse tumor caliper measurements were calculated in tumor volume (V) using the formula:

$$V = 1/2(\text{Length} \times \text{Width}^2) \qquad (1)$$

## Mouse blood collection and plasma preparation

70–100 μl of blood was collected from the mouse facial vein using Microvette 100 tube (Cat:20.1282.100, SARSTEDT). Plasma was collected by centrifuging the mouse blood for 20 min at $2000 \times g$ via a refrigerated centrifuge. Once the plasma was collected, all samples were stored at −80 °C until use. Two criteria established the design of sample collection. First, samples were collected until a significant increase in tumor size could be physically seen growing on the mice. Second, samples were collected until no major increase was seen on the sEV count detected by PANORAMA.

## Ethical regulations for animal testing

We adhered to ethical regulations for animal testing involving live vertebrates. Ethical approval was granted at UT MD Anderson Cancer Center under the IACUC protocol 00001137-RN03. This protocol was approved on 9/8/22 and is valid until 9/7/25. Animals used were nude female mice (0002019 NU/J, Jackson Laboratory), 8-week-old and weighing around 25 g each.

## Procedure for plasma extraction from healthy and cancer participants and PANORAMA analysis

All participants have consented to institutional review board at MD Anderson Cancer Center-approved biomarker protocols. Healthy volunteers have consented to PA14-0063, and the one-time blood sample was collected and processed in the laboratory. Cancer patients were also consented to institutional review board at MD Anderson Cancer Center approved prospective biomarker or therapeutic protocols, depending on the disease site. Blood was collected through venipuncture before treatment started (radiation, surgery or chemotherapy) and the extracted plasma was aliquoted and stored in −80 °C until use. The utilization of these samples for sEV analysis was done on protocol 2021-0368, which allows the analysis of de-identified samples collected from consented patients on IRB-approved protocols. The approved IRB ID is 2021-0368_CR001; registration ID IRB 2 IRB00002203. To distinguish the sEV population in the pre-wash sample, a PANORAMA contrast threshold was established at 18% which translates to 200 nm which is the upper size limit of exosomes. This number was established based on the naturally occurring size of sEV derived from both experimental results and previously reported size ranges in literature. All detected particles post-wash were particles bound to the functionalized surface and are termed retained sEVs. Further analysis of retained sEVs using fluorescence imaging allowed the identification of *miR-21* within the population. *miR-21* occurrence was reported in percentage values which was the number of sEVs with positive *miR-21* among the total number of retained sEVs.

## Molecular beacons

A molecular beacon probe complementary to *miR-21* labeled with Cy3 was used to probe for intravesicular micro-RNA content (5Cy3/GCGCGTCAACATCAGTCTGATAAGCTACGCGC/3BH, Integrated DNA technologies). The probe contains the complementary sequence to *miR-21* and has complementary arms that form a stem-loop (hairpin) structure when they hybridize with one another. The hairpin structure places the Cy3 near the fluorescence quencher BH which prevents any fluorescence when not hybridized with the target sequence (*miR-21*). The sEVs were incubated in a water bath set at 37 °C for 2 h with the molecular beacon (10 nM). The heating facilitates the molecular beacon to enter the sEVs and unwinds its hairpin structure to hybridize with the target sequence. Removing the sEV-MB solution from the heat allows non-hybridized probes to revert to the hairpin structure. Any probe hybridized to the target sequence remains in "straight" form which places Cy3 and BH far from one another and permits fluorescence emission[19].

## PANORAMA

With the trans-illumination geometry, the camera receives transmitted light passing through the AGNIS after a bandpass filter. The bandpass filter is located near the half-max wavelength on the left shoulder of the AGNIS's LSPR extinction curve. When an imaging target resides outside the AGNIS's longitudinal sensing range, i.e., too far away from the surface, it will show up as a standard light-scattering object with its intensity reduced when transmitted through the AGNIS. However, as the imaging target approaches AGNIS, the elevated local RI causes the LSPR extinction curve to redshift. The LSPR redshift causes increased light transmission within the imaging wavelength range and acts as if a virtual nano-aperture is formed right beneath the target. We emphasize that the nano-aperture allows higher transmission for both the nanoparticle scattered light and the unscattered incidence light. In other words, unlike most scattering-based imaging techniques, PANORAMA relies on unscattered light to detect nanoparticles smaller than the imaging resolution[13]. PANORAMA images are obtained as intensity ratio (IR) by dividing an image with a target-free background image. The IR value further provides size information of the detected particle. Alternatively, PANORAMA contrast, defined as (IR-1)%, can be calculated and subsequently employed in this paper. Under our current instrument conditions, background contrast is <1.5% (mean+3*std).

## Optical setup

**PANORAMA.** White light from a tungsten-halogen lamp passes through a condenser (IX-LWUCD, Olympus) and illuminates the sample on an inverted microscope (IX71, Olympus). The transmitted light passes through an infinity-corrected 60X water immersion lens with a 1.2 numerical aperture (UPLSAPO60XW, Olympus). The light exiting the side-port is relayed to an electron multiplied charge-coupled device (EMCCD; ProEM 1024, Princeton Instruments) via a 4f system, where a bandpass filter with 650–670 nm passband (FB660-10, Thorlabs) is at its fourier plane. sEVs PANORAMA measurements were done in PBS 1X.

**Fluorescence.** A blue collimated light source (Thorlabs, M470L4-C1) was used for the excitation of PKH67 labeled sEVs. Emitted fluorescence signals were collected with a 465/−490-/500-/525-nm filter set (Thorlabs). The emitted fluorescence light is relayed onto the EMCCD using the same 4 f system used for PANORAMA. The exposure time of the EMCCD was set at 100 ms for sEVs. A Yellow/Orange collimated light source (Thorlabs, MINTL5) was used for the excitation of Cy3 labeled *miR-21* molecular beacon. Emitted fluorescence signals were collected with a 512/−550-/570-/615-nm filter set (Edmund Optics).

## Image processing

**PANORAMA.** PANORAMA images were aligned with a target-free background image using the auto-alignment program in ImageJ. After the alignment, the PANORAMA images were divided by background images resulting in an image with a mean value of 1 and a standard

deviation of 0.005. threshold IR value was selected to be $1 + 3 \times 0.005 = 1.015$ (1.5% contrast). Assuming Gaussian statistics, a contrast value larger than 1.5% indicates particle detection. Particle counting was done by determining the local maxima of each detected particle and avoiding counting the particle twice. This was done using the find maxima function in ImageJ. The contrast of the local maxima was used as the contrast of each particle.

**Fluorescence.** Fluorescence images were also aligned with a target-free background image using the auto-alignment program in ImageJ. After the alignment, the fluorescence images were subtracted by background images resulting in an image with a mean value of 0 and a standard deviation of 10. The threshold fluorescence value was selected to be 30 (mean+3*std). Assuming Gaussian statistics, a fluorescence value larger than 30 indicates particle detection with *miR-21*.

### Statistical methods

**Receiver-operating characteristic (ROC) analysis.** In this study, ROC analysis was employed to assess the performance of our diagnostic test and evaluate the accuracy of our predictive model. The cutoff for PANORAMA-derived small extracellular vesicles (sEV) count was systematically varied across a range from 2 to 1682, using an incremental step size of 2. The True Positive Rate (TPR, sensitivity) and False Positive Rate (FPR, complementary facet of specificity) were computed based on these cutoff variations, generating the ROC curve. The Area Under the ROC Curve (AUC) was determined to be 96.86%.

The formula to calculate TPR and FPR are given as follows:

$$\text{TPR} = \text{Sensitivity} = \frac{True\ Positive}{True\ Positive + False\ Negative} \quad (2)$$

$$\text{FPR} = 1 - \text{Specificity} = \frac{False\ Positive}{False\ Positive + True\ Negative} \quad (3)$$

**P value.** The *p*-value is based on the following formula:

$$z = \frac{p - p_0}{\sqrt{\frac{p_0(1-p_0)}{n}}} \quad (4)$$

Where:

$p$ represents the sample proportion,

$p_0$ signifies the assumed proportion stipulated by the null hypothesis, and n corresponds to the sample size.

Upon obtaining the value of z through this computation, the associated *p*-value is obtained by referencing a z-table.

### Reporting summary

Further information on research design is available in the Nature Portfolio Reporting Summary linked to this article.

## Results and discussion

### Digital sEV counting by PANORAMA in conjunction with fluorescence imaging

We conjectured that the counting and analysis of sEVs in a digital fashion could provide greater sensitivity and dynamic range in sEV detection. To first confirm that the nanoparticles captured on the antibody-functionalized AGNIS surface are indeed sEVs, primarily consisting of exosomes, we performed time-lapsed PANORAMA monitoring of sEV settlement and binding on AGNIS until an endpoint at ~125 min, followed by a washing step to remove non-specific binding. Fluorescence images were also acquired at the endpoint as well as after the washing step. The antibody binding specificity was determined using non-specific IgG antibodies. Overall, only 3% of detected sEVs before wash were retained on the IgG-functionalized AGNIS surface after wash (Supplementary Fig. S1). The sEVs in this experiment were ultracentrifugation (UC) purified from H460 lung

cancer cell culture supernatant and labeled using PKH67 fluorescent membrane lipophilic dyes. The AGNIS surface was functionalized with antibodies specific for exosomes surface protein CD63. Figure 1a shows the time-lapsed PANORAMA images during sEV settlement, revealing their gradual accumulation over time (Fig. 1a). This accumulation of sEVs serves as a built-in quality control mechanism for robust sEV detection and counting. The number of settled sEVs within these images is summarized in Fig. 1b, showing a gradual increase in sEV counts over time, peaking at 125 min, and followed by the anticipated reduction in sEV counts after the washing step (Fig. 1a, b). Figure 1a also includes full field-of-view PANORAMA images of total detected sEVs before (count 3133) and after wash (count 2910) indicating a 93% retention of settled sEVs. Fluorescence images of PKH67 membrane labeling dye were also taken at each step and timepoints, demonstrating complete agreement with the PANORAMA images, indicating that PANORAMA-detected particles were sEVs. While it is possible that high-density protein aggregates sedimented by ultracentrifugation could be visualized by PANORAMA, PKH67 fluorescence was observed in all PANORAMA detected particles before and after washing, suggesting that the extracted sEVs were of high purity and absent of any high-density protein aggregates. As demonstrated previously[13], PANORAMA contrast correlates well with nanoparticle size. Similar behavior is also expected from PKH67 fluorescence intensity, assuming the dye distributes evenly on the sEV membrane. To examine whether they provide consistent size information, fluorescence intensity is plotted against PANORAMA contrast at the single sEV level (Fig. 1c). We found PANORAMA contrast to linearly correlate with fluorescence intensity (Pearson correlation +0.72, $p < 0.001$). Figure 1d shows the contrast histogram of all detected sEVs both before and after wash, with average contrast of $11.8\% \pm 5.1\%$ and $11.5\% \pm 4.5\%$, respectively (Fig. 1d). The washed away sEVs showed a contrast range of 3–40%, with an average contrast of 17.5% (±10%). This suggests that the washed away particles are of comparable size to the AGNIS bound sEVs, although slightly larger on average, and can still be considered as sEVs without the CD63 protein expression.

### sEV sizing via PANORAMA

PANORAMA's sizing capability originates from the quantitative relationship between the LSPR shifts and local refractive index changes. As we have reported previously, PANORAMA was capable of distinguishing polystyrene bead (PSB) size based on the contrast[13]. For example, the contrast values for 200, 100, 50, and 25 nm PSB were 21%, 17.5%, 14%, and 10%, respectively. The contrast-based sizing capability is crucial because most sEVs are much smaller than the diffraction limit and the sampling resolution of our imaging system. Size information cannot be obtained reliably by measuring the spot area on the camera. Herein, we explored whether PANORAMA could generate size distribution for sEVs. Due to the significantly lower refractive index of sEVs, PSB cannot be used for size calibration. Instead, we employed the size distribution obtained from Nanosight to calibrate PANORAMA contrast and obtained a contrast-to-size conversion relationship:

$$Y = 0.1531x^2 + 7.096x + 20.211, \quad (5)$$

where Y (nm) is the size and x (%) is the PANORAMA contrast. This was further evaluated on a separate set of H460 cancer cell line sEVs. Figure 1e shows the size distribution characterized by Nanosight and PANORAMA, where Nanosight reports an average size of $135 \pm 102$ nm with the smallest particle size at 48 nm while PANORAMA provides an average size of $122 \pm 48$ nm with the smallest being 34 nm. The majority of the sEVs are mostly well within the 50 to 200 nm range using both sizing methods, which is the target size range to encompass as many exosomes as possible in the sEV population. However, PANORAMA shows superior sensitivity and sizing resolution for sEVs smaller than 75 nm compared to Nanosight. Furthermore, to capture the size distribution of whole vesicle population isolated by UC, we utilized PANORAMA to detect the vesicles on an IgG

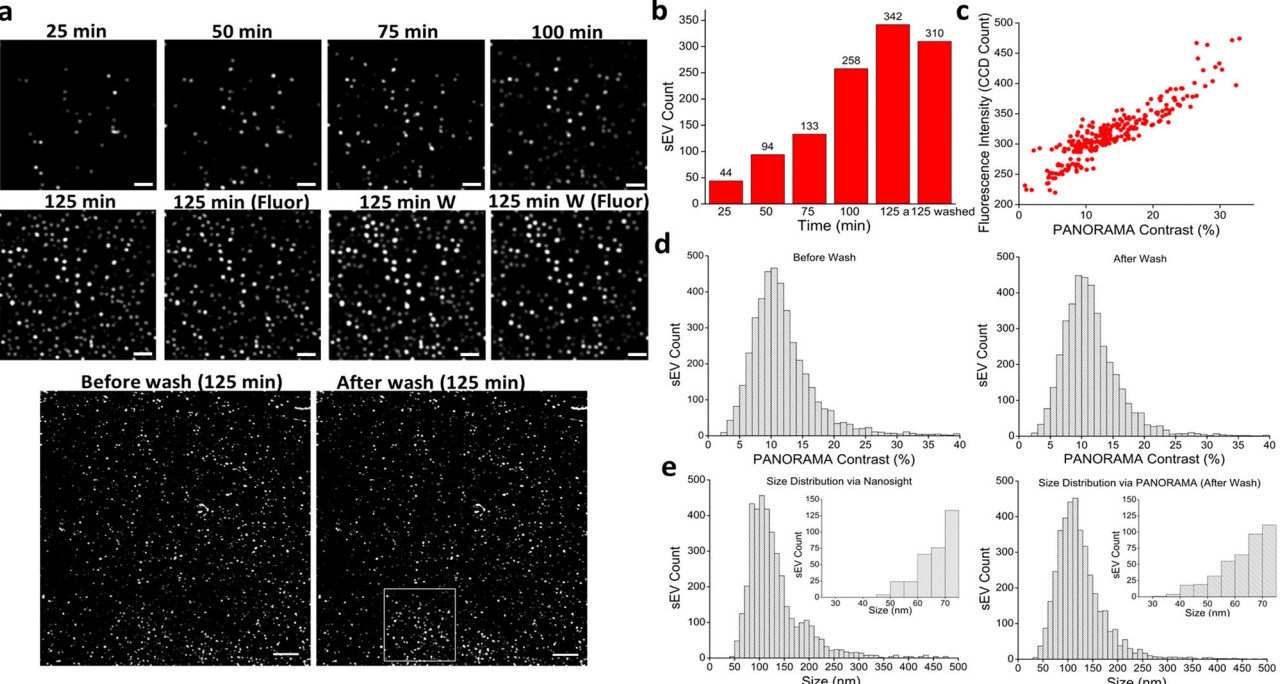

**Fig. 1 | Digital counting and sizing of sEVs via PANORAMA. a** Time-lapsed PANORAMA images of H460 sEVs at times 25 min, 50 min, 75 min, 100 min, and 125 min before wash, and after wash (indicated with W), fluorescence images (indicated with Fluor) at time 125 min before and after wash (indicated with W), and PANORAMA images at the full field of view showing all detected sEVs before and after wash alongside the selected region chosen for time-lapsed images. **b** sEV counts vs. time from PANORAMA images in (**a**). **c** Fluorescence intensity vs. PANORAMA contrast of sEVs detected after wash. **d** PANORAMA contrast histogram of total detected H460 sEVs before and after wash. **e** Size distribution of H460 sEVs via Nanosight and via PANORAMA. Scale bar: (**a**), 2 μm for selected region images; 10 μm for full field of view images. PANORAMA Plasmonic nano-aperture label-free imaging, sEV small extracellular vesicles.

functionalized surface before washing. We applied the same calibration curve used for Nanosight measurements to determine the vesicle size based on PANORAMA contrast (Supplementary Fig. S1a and S1b). The average PANORAMA contrast before washing was measured to be 12.54 ± 5.3%, corresponding to a size of 133 ± 56 nm (size distribution shown in Supplementary Fig. S1c). The size distribution of vesicles detected via Nanosight (135 ± 102 nm) and PANORAMA on the IgG functionalized surface (133 ± 56 nm) represents the same whole vesicle population obtained through UC. Notably, the size distributions obtained from both techniques exhibit excellent agreement, demonstrating the robustness of the calibration curve. This indicates that the calibration curve can be reliably applied to interpret the contrast distribution of sEV across different experiments conducted using PANORAMA.

## Quantifying sEV cargo using molecular beacon probes

While sEVs contain protein, lipids, and nucleic acids that reflect the molecular content of the cell of origin, sEV cargo molecules such as *miRs* have been recognized as potential cancer biomarkers[7]. To go beyond single sEV enumeration, we have selected *miR-21*, a well-established cancer biomarker[20,21]. It is important to note that utilizing plasma *miR-21* is conceptually similar to detecting CTDNA, which can offer high specificity and sensitivity for cancer detection. However, there are notable limitations when using CTDNA for cancer screening. The lack of early-stage DNA shedding poses challenges for early diagnosis. The low abundance of CTDNA in the bloodstream and its susceptibility to degradation further complicate reliable detection and quantification. Variability in CTDNA levels over time and their dependency on factors like tumor size and treatment introduce difficulties in obtaining consistent results. The cost and technical requirements of advanced technologies like next-generation sequencing can limit the accessibility of CTDNA analysis. Furthermore, plasma *miRs* are prone to be degraded by nucleases rather than being protected in sEVs. In contrast, sEVs offer advantages in their abundance and capacity of storing genetic material for extended periods, making them good candidates for biomarker analysis.

Therefore, utilizing sEV *miR-21* could potentially provide enhanced sensitivity and reproducibility to detect cancer at early stages, all at a significantly lower cost.

For sEV *miR-21* analysis, we employed a molecular beacon (MB) probe containing a 3' BHQ2 quencher molecule that neutralizes the 5'-Cy3 fluorophore that would fluoresce when hybridized with intravesicular *miR-21*. Purified sEVs from various cell lines ($2 \times 10^3$ sEVs/μl) were incubated with the *miR-21* MB before placing them on the AGNIS chip. To maximize the binding efficiency of exosomes with the purified sEVs population expressing differing levels of surface tetraspanin proteins, we functionalized the AGNIS surface with a cocktail of anti-CD9, anti-CD63, and anti-CD81 antibodies[22–24]. The gradual settling and accumulation of sEVs at intervals over 60 min and the number of sEVs retained on the AGNIS chip after wash was monitored using PANORAMA (Fig. 2a). For H460 sEVs, before and after wash showed 95% (±1.2%, N = 4) retention on the chip. We localized the Cy3-labeled sEVs over the PANORAMA image and found 79% (±7.7%, N = 4) of the H460 sEVs were *miR-21* + .

We applied the same approach to quantify both the retention fraction and the percentage of sEVs containing *miR-21* to three "non-cancerous" cell lines: 293 A, MCF10A, and NHBE. The average retained fraction after wash for sEVs derived from 293 A, MCF10A, and NHBE cell lines were 87% (±3.4%, N = 4), 86% (±2.7%, N = 4), and 79% (±7.2%, N = 5), respectively. Overall, the retention percentage was consistent among the different non-cancerous cell line samples (p > 0.05), suggesting that the functionalized surface had a general affinity to retain sEVs regardless of the cell of origin. This is expected, as the tetraspanin proteins are expressed on exosomes regardless of their origin. This serves as an indicator that the majority of the detected sEVs consist of exosomes. In contrast to H460 sEVs of which three-quarters contain *miR-21*, sEVs derived from non-cancerous cell types displayed a lower *miR-21* occurrence (14.6% ± 1.15%, N = 4, 8.5% ± 1.5%, N = 4, and 0%, N = 5; all p < 0.01). Although non-cancerous, sEVs derived from 293 A and MCF10A cell lines are abnormal likely due to their immortalized condition, which could be the reason for the lower *miR-21*

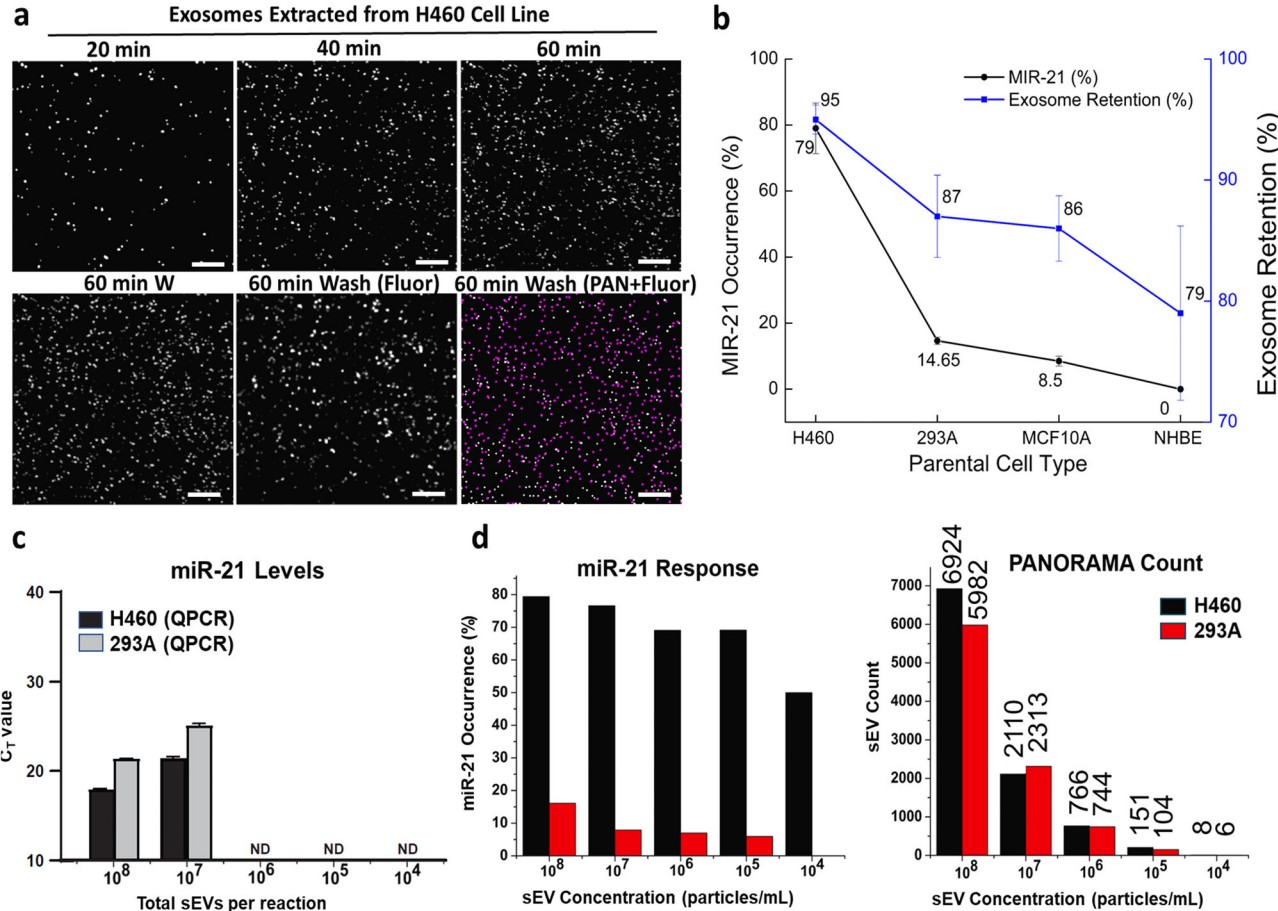

**Fig. 2 | PANORAMA digital counting and cargo *miR-21* profiling.**
**a** PANORAMA images of H460 sEVs settled at 20 min, 40 min, 60 min, and 60 min post-wash alongside fluorescence images of sEVs at 60 min post-wash, and binarized image of overlayed post-wash PANORAMA and fluorescence, where the white spots are sEVs detected only by PANORAMA and magenta spots are sEVs detected by *both* PANORAMA and fluorescence (indicating *miR-21* positive). **b** *miR-21*

occurrence (left axis) and sEV retention % (right axis) vs. parental cell types. **c** *miR-21* occurrence at various concentrations of H460 & 293 A sEVs using qPCR. **d** *miR-21* occurrence and detected sEV count at various concentrations of H460 & 293 A sEVs using PANORAMA. Scale bar: 10 μm. Error bars in (**b**, **c**) capture the range of variability observed across multiple repetitions of the study. PANORAMA Plasmonic nano-aperture label-free imaging, sEV small extracellular vesicles.

expression. However, sEVs derived from non-immortalized NHBE cells had no *miR-21* expression, which points to the potentially high specificity of *miR-21* for immortalized or transformed cells (Fig. 2b).

We further conducted a comprehensive comparison of our findings using quantitative polymerase chain reaction (qPCR) and PANORAMA, where experiments involved varying concentrations of sEVs derived from H460 and 293 A cell lines. The qPCR results indicated a 7.6-fold difference in $C_T$ values between H460 and 293 A sEVs at the highest concentration tested ($1 \times 10^8$ particles/reaction). The lowest detectable occurrence of *miR-21* was observed at a concentration of $1 \times 10^7$ particles/reaction from both cell lines (Fig. 2c). On the other hand, PANORAMA experiments were conducted using 20 μL of sEV solutions at different concentrations, mirroring the concentrations used in qPCR. Figure 2d shows that the lowest detectable *miR-21* occurrence for H460 was at a concentration of $1 \times 10^4$ particles/mL (equivalent to 200 particles/experiment), while for 293 A, it was at a concentration of $1 \times 10^5$ particles/mL (equivalent to 2000 particles/ experiment). The *miR-21* occurrence for H460 varied from 50 to 79% for all different concentrations, and for 293 A, it varied from 6 to 16% across all concentrations. Notably, PANORAMA demonstrated the capability to detect and quantify sEVs at a concentration as low as $1 \times 10^4$ particles/mL from both cell lines, corresponding to a limit of detection (LOD) of 16.7 attoM for sEVs (Fig. 2d). These results highlight the enhanced sensitivity of PANORAMA in comparison to qPCR, particularly in the detection of *miR-21* occurrence at lower concentrations. Furthermore, the exact subset of

*miR-21*-positive sEVs can be examined at the single unit level by the combination of PANORAMA and fluorescence imaging.

### Longitudinal monitoring of cancer sEV counts in mouse plasma
To determine if cancer cell-derived sEVs could be directly detected and quantified from plasma without purification, we performed a series of experiments on human tumor xenograft in mice (Fig. 3a). H460 cells expressing GFP-tagged CD63 (SBI, Palo Alto, CA) were implanted into nude mice at the leg region to minimize mouse health complications, allowing the tumor to form over time. About 20–50 μl of plasma was extracted from 50 to 100 μl blood collected in EDTA tubes was drawn from each mouse once a week. 20 μl plasma was analyzed using the same PANORAMA-Fluorescence imaging protocol described earlier with the same AGNIS substrates functionalized with the triple tetraspanin antibody cocktail. Plasma samples from control mice without tumor implant showed on average 8 ± 5 retained particles with no *miR-21* or GFP-labeling. This is likely the non-specific binding of debris from plasma that is retained on the AGNIS substrate. Similarly, baseline plasma drawn from the cancer cell injected (CCI) group (mice 113–118) showed a similar particle count range as the control group (Fig. 3a). However, these animals were tracked over time, and once tumor size reached 4 mm³, particle counts would begin to increase with a fraction of the particles to be *miR-21*+ and *GFP*+ (Fig. 3b). This was seen for all CCI mice except for mouse 116 which never developed a tumor and continued to show minimal counts of retained particles without

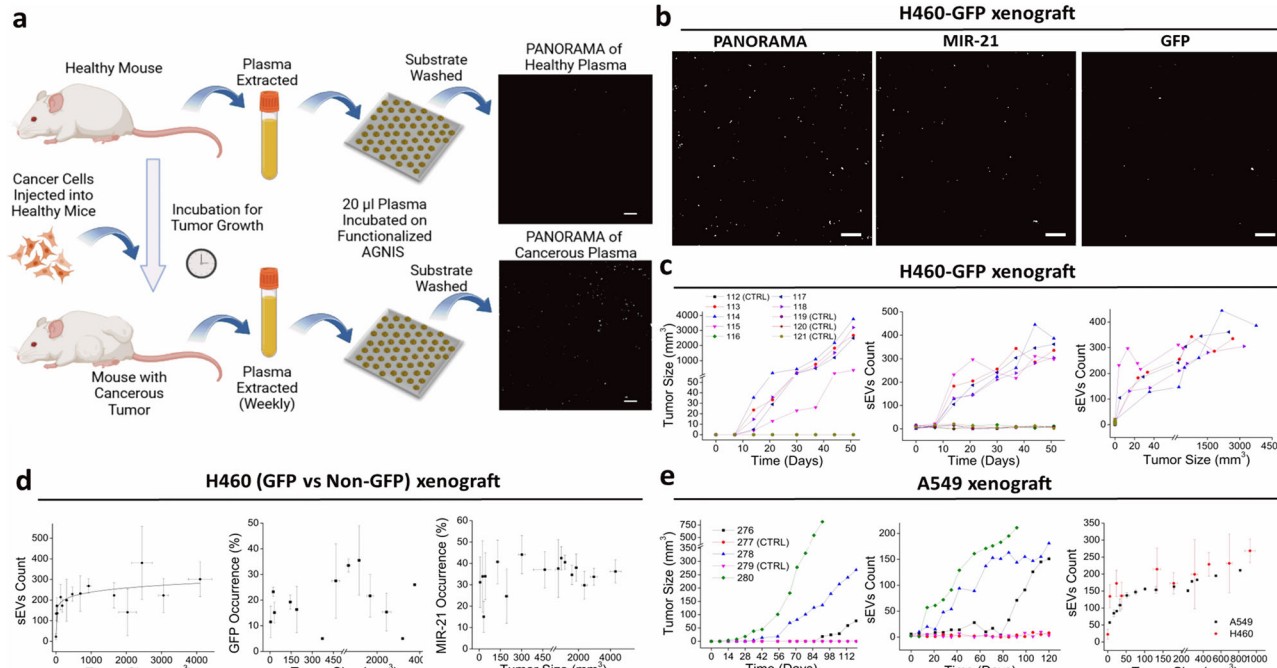

**Fig. 3 | Longitudinal monitoring of sEVs in mouse plasma. a** Experimental procedure to count and profile sEVs from mouse plasma. **b** PANORAMA image of H460 (*GFP*) sEVs alongside fluorescence image showing *GFP* and *miR-21* occurrence. **c** Longitudinal monitoring of tumor size, plasma sEV counts (post-wash), and sEV counts post-wash vs. tumor size. **d** Average sEVs count (post-wash), *GFP*, and *miR-21* occurrence vs. tumor size for H460 CCI mouse plasma (*n* = 10, 10 samples extracted). **e** Longitudinal monitoring of tumor size, sEV counts (post-wash), and sEV counts (post-wash) vs. tumor size for A549 CCI mouse plasma. sEV counts

(post-wash) vs. tumor size for H460 CCI mouse plasma are reported for comparison (*n* = 5, 16 samples extracted). Scale Bare: 10 μm. The error bar in graph (**d**) reflects the variability of retained sEV count and the percentage of *GFP* and *miR-21* occurrences within the retained sEVs across various tumor sizes originating from H460 cell lines. The error bar in (**e**) reflects the variability of retained sEV count across different tumor sizes originating from A549 cancer cells. PANORAMA Plasmonic nano-aperture label-free imaging, sEV small extracellular vesicles.

*miR-21* or *GFP* occurrence. Furthermore, within two weeks of the cancer cell injection, all CCI mouse plasma samples exhibited a minimum of a 10-fold increase in post-wash particle count compared to the control values and a slight increase in the percentage of retained particles with *miR-21* and *GFP* labeling (Fig. 3c and Supplementary Fig. S2). Interestingly, the particle count does not increase linearly with the tumor size. The increased rate of absolute sEV counts (ASC) post-wash in all CCI mice reaches a plateau value of about 234 sEVs (±61) (11.7 ± 3 ASC/μl) when the tumors reached 150 to 200 mm³. Comparable results in particle retention and *miR-21* positivity are seen with the parental H460 (non-*GFP* labeled) lung cancer cells (Supplementary Fig. S3). A power law is employed to capture the relationship of retained sEVs and tumor size:

$$Y = 101.7|x - 3.48 \times 10^{-7}|^{0.12013} + 5.42, \qquad (6)$$

where Y is the sEV count post-wash and x (mm³) is the tumor size where the earliest tumor size can be measured at an average of 4.94 mm³ ±1.15 mm³ (Fig. 3d). Similar to the sEV counts, *GFP* and *miR-21* appeared at similar proportions regardless of the tumor size. On average, 21% (±11.4) and 28% (±11.5) of sEV population exhibited *GFP* and *miR-21* occurrence, respectively (Fig. 3d). This behavior of *miR-21* occurrence can also be seen for H460 (non-*GFP*) CCI mice with an average of 36 % (± 12%) (*p* > 0.01) (Supplementary Fig. S3).

To determine if these results could be replicated in another cell line with differing growth kinetics, we examined this in the A549 lung cancer line (Fig. 3e). Much like the H460 cells, all mice that formed a palpable tumor would begin to show a detectable increase in plasma sEV counts. One of the mice, #276, did not exhibit any increase in sEV count above baseline until day 80 after injection which was the time it took for the mice to develop a palpable tumor. Similar to H460, A549 CCI also displayed similar plateauing in sEV counts as the tumor reaches a certain level beyond which the

count increase slows down significantly. These results suggest that sEVs excreted from cancer cells into the blood can be readily detectable by PANORAMA even when the tumors are very small. However, sEVs do not linearly increase with tumor size, but rather reach a steady state level after tumors reach a certain size threshold (Fig. 3d). This observation has also been seen in other studies using plasmonic and ELISA-based assays[25,26].

### Digital sEV counting from human plasma to differentiate healthy and cancer participants

Based on what we have learned from the animal studies, we aimed to apply this approach to human samples by directly quantifying sEVs captured from human plasma. We took a set of healthy donor plasma (HDP) and cancer patient plasma (CPP) samples and applied 20 μl of each sample on the functionalized AGNIS surface and monitored over 60 min. Figure 4a shows the images of particles detected via PANORAMA at 60 min (pre- and post-wash) and fluorescence (post-wash) from a representative HDP sample (time-lapsed PANORAMA images are in Supplementary Fig. S4). The number of settled particles indicated a gradual increase reaching an equilibrium level of 145 at 60 min. Based on sEV contrast distribution in Fig. 1e, particles with PANORAMA contrast of up to 18% are classified as "sEV-like", while the remaining are considered large non-sEV entities which may include microvesicles and cell blebbing. The contrast cutoff for sEV-like particles at 18% translates to ~200 nm using the contrast-to-size conversion as we have shown previously (Fig. 1)[1,7]. After washing away the non-specifically bound particles, the sEV count was reduced to 37 with a contrast average of 9.3% ± 3.4%, an average size of 99 ± 36 nm, and a retention percentage of 36.6%. The size distribution of sEVs (post-wash) from HDP ranged from 50 to 150 nm, a range well comparable to the size range of sEVs from the H460 cell line (Fig. 1f) and those reported in the literature for exosomes (50–200 nm) (Supplementary Fig. S4). No *miR-21* labeling was detected. Next, we compared these results to a sample from a representative

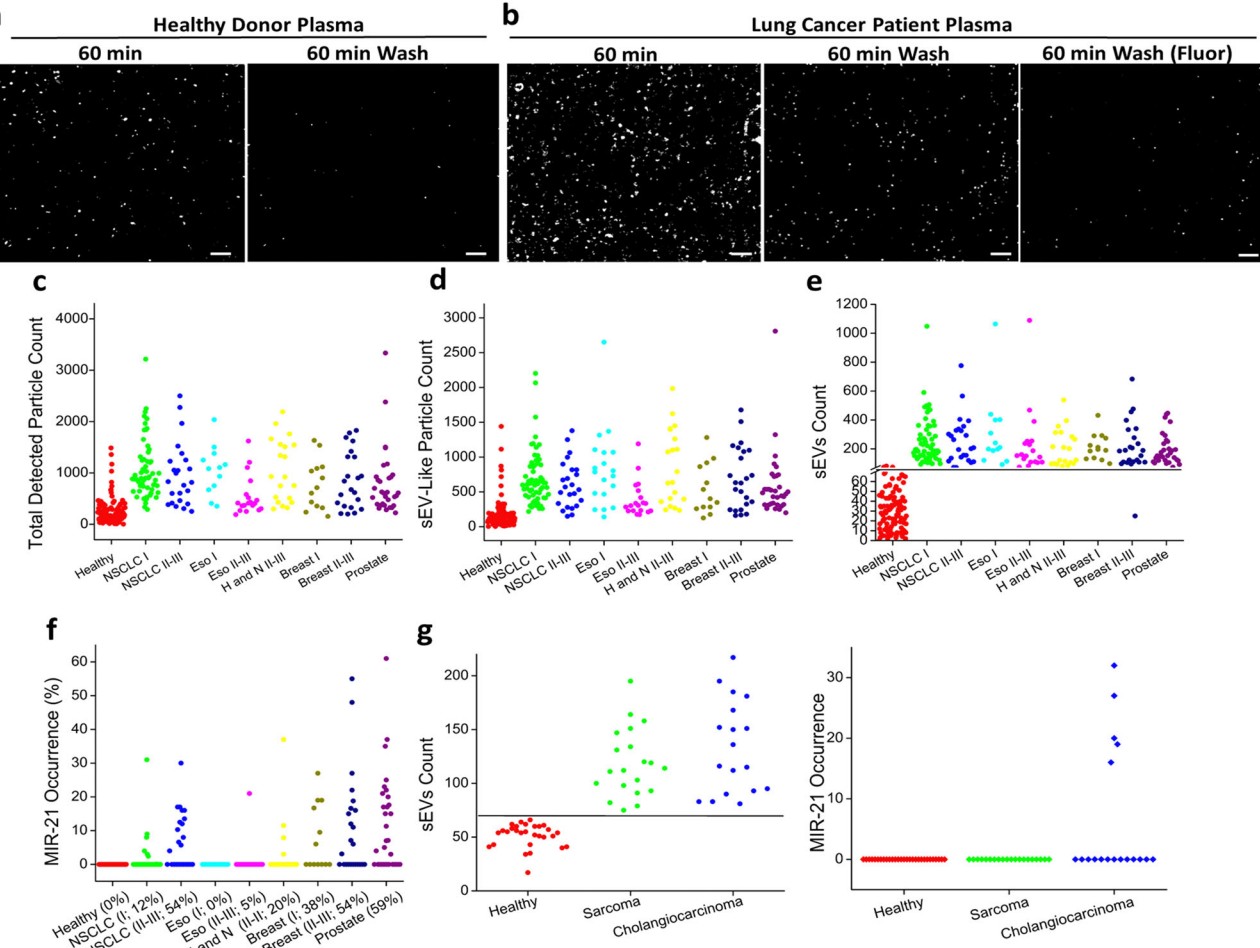

**Fig. 4 | Profiling plasma sEVs from healthy donors and cancer patients. a** Pre- and post-wash PANORAMA images of a representative healthy donor's plasma sample. **b** Pre- and post-wash PANORAMA images and post-wash fluorescence images of a representative lung cancer patient's plasma sample. **c** Total detected particle counts (pre-wash), (**d**) sEV-like particle count (pre-wash),(**e**) sEV count (post-wash), and (**f**) *miR-21* occurrence among sEVs (post-wash) of healthy donor plasma and different cancer plasmas obtained via PANORAMA-Fluorescence imaging system. **g** sEV count (post-wash) and *miR-21* occurrence within sEVs (post-wash) of blind tested Sarcoma and Cholangiocarcinoma cancer patient plasma. Scale bar: 10 μm. PANORAMA Plasmonic nano-aperture label-free imaging, NSCLC non-small cell lung cancer, Eso esophagus, H and N head and neck, sEV small extracellular vesicles.

stage III non-small cell lung cancer (NSCLC) CPP sample collected at baseline before treatment. Figure 4b shows the images of particles detected via PANORAMA at 60 min (pre- and post-wash) and fluorescence (post-wash) and time-lapsed PANORAMA images (Supplementary Fig. S5). Total settled and sEV-like particle counts indicated a gradual increase reaching a maximum count of 1522 and 1068 at 60 min, respectively. Post-wash sEV count was 302 with a retention percentage of 28.2% an average contrast of 9.0% ± 3.1%, an average size of 96 ± 33 nm, and a size distribution ranging from 50 to 150 nm (Supplementary Fig. S5). The fluorescence image showed 8% *miR-21* occurrence (Fig. 4b).

Based on the results from the mouse models and these initial samples from human participants, we conjectured that sEV concentration could also reach a higher steady-state level compared to non-cancer individuals and would not vary significantly by tumor stage since the steady-state level would have been reached even at the earliest stage of tumor development. To test this hypothesis, we examined samples from an extensive set of healthy donors and cancer patients (Fig. 4a, b). Multiple CPP samples were collected from patients with non-metastatic NSCLC (stage I, $N = 60$; stage II–III, $N = 24$), esophageal cancer (stage I, $N = 13$; stage II–III, $N = 19$), head and neck cancer (stage II–III, $N = 20$), breast cancer (stage I, $N = 13$; stage II–III, $N = 24$), and prostate cancer ($N = 32$). A series of additional HDP samples were analyzed to serve as controls ($N = 106$). The evaluators were blinded to the identity of the samples and reported objectively the particle counts before and after wash, and the proportion of fluorescently labeled particles (Fig. 4c, d, e, and Supplementary Data 2). HDP samples showed an average total detected particles of $242 \pm 254$ ($12.05 \pm 12.7$ ASC/μl) and sEV-like particles of $178 \pm 214$ ($8.9 \pm 10.7$ ASC/μl) before wash, and $30 \pm 19$ ($1.5 \pm 0.95$ ASC/μl) retained sEVs without any *miR-21* occurrence after wash. The cancer subgroup plasmas showed an average total detected particle of $947 \pm 632$ ($47.5 \pm 31.6$ ASC/μl) and sEV-like particles of $664 \pm 440$ ($33.2 \pm 22$ ASC/μl) before wash, and retained sEV count of $244 \pm 191$ ($12.2 \pm 9.6$ ASC/μl) after wash. While there were 8-fold higher level of retained sEVs in the cancer samples ($p < 0.001$), there were no statistically significant differences in the levels of sEVs comparing the different tumor types or by the stage of cancer, both within and between tumor types ($p > 0.1$). *miR-21* was only detected in the cancerous plasma sEVs, but not all cancer patient plasmas within each subgroup expressed *miR-21* (Fig. 4f). *miR-21* was detected within 12 % and 54% of analyzed plasmas with stage I and stage II NSCLC patients, respectively. Esophageal cancer plasma did not show any presence of *miR-21* within stage I patient but showed 5% for stages II-III. *miR-21* was detected within 20% of tested plasma with head and neck cancer (II–III), 38% and 54% of plasma with stage I and stage II–III breast cancer patients, respectively, and 59% in prostate cancer. Breast and prostate cancers showed the highest *miR-21* occurrence among the analyzed CPP samples, consistent with what has been detected in tumor tissues[27–31].

Notably, even before the washing step, CPP samples showed higher total sEV-like particles compared to HDP. However, relying on pre-wash sEV count does not seem to distinguish CPP from HDP samples with high performance. In addition, a slight increase in the retention percentage is observed in CPP samples but not statistically significant ($p > 0.1$) (Supplementary Fig. S6a). In contrast, post-wash sEV counts from CPP samples were significantly higher compared to HDP and can consistently separate the CPP and HDP samples ($p < 0.01$) (Fig. 4c and Supplementary Fig. S6b). Using a cutoff of 70 (3.5 ASC/µl), the post-wash sEV counts for all CPP samples ($N = 205$) were above this threshold except for one sample, and for HDP samples ($N = 106$), all but three were below this cutoff, giving a cancer detection sensitivity of 99.5% and specificity of 97.3%. The Receiver-operating characteristic (ROC) analysis is provided in the Supplementary Fig. S7 with an area under the curve (AUC) of 96.86%.

To validate the performance of the diagnostic threshold of 70 retained sEV counts in 20 µl plasma (3.5 ASC/µl), we analyzed two independent sets of samples from stage I–IV or recurrent leiomyosarcoma/gastrointestinal stromal tumors ($N = 20$, labeled as sarcoma in Fig. 4) and early- and late-stage cholangiocarcinoma ($N = 18$) that were anonymously labeled and mixed in with HDP samples ($n = 29$). HDP samples showed a post-wash average of $51 \pm 10$ ($2.6 \pm 0.5$ ASC/µl) retained sEV count with no values above 70 and no *miR-21* expression. In contrast, CPP samples showed an average of $118 \pm 30$ ($5.9 \pm 1.5$ ASC/µl) and $133 \pm 42$ ($6.7 \pm 2.1$ ASC/µl) for sarcoma and cholangiocarcinoma, respectively, without values below 70 (3.5 ASC/µl) (Fig. 4g). While the sarcoma plasma samples did not show any *miR-21* labeling, 5 of the cholangiocarcinoma plasma samples showed *miR-21* occurrence (Fig. 4g).

## Discussion

PANORAMA in this study serves as a label-free, unbiased way for detecting, sizing, and quantifying sEVs in plasma samples. PANORAMA facilitated the visualization and enumeration of sEVs based on their endogenous properties, eliminating the need for exogenous labels or modifications. By utilizing PANORAMA, we captured real-time images of sEVs and analyzed their characteristics, such as size, providing valuable insights into the sEV population in the samples. Furthermore, the distinct sEV levels quantified by PANORAMA from diseased and healthy populations have been clearly observed in animal models and human participants. It's important to note that sEVs count may also be elevated with non-cancer conditions, such as inflammatory conditions or chronic diseases, an area ripe for future research. Therefore, the PANORAMAbased protocol could be used as a sensitive initial cancer screening technique, followed by additional diagnostic tests to confirm the nature of cancer using specific biomarkers such as *miR-21* and many others.

Fluorescence played a crucial role in this study by specifically detecting and quantifying the presence of *miR-21*, an example of one well-established cancer biomarker, within the identified sEVs. Fluorescence imaging enabled the selective labeling and visualization of *miR-21* within the PANORAMA-detected sEV population, providing additional information about the presence and abundance of this specific biomarker. It is important to note that not all cancers express the same genetic biomarkers. Moreover, the occurrence of *miR-21* in plasma samples from different cancer subsets has shown variations. Thus, relying solely on *miR-21* for cancer screening may have limitations and lack the sensitivity to detect all types of cancer, potentially being restricted to a few specific types.

Overall, the combination of PANORAMA and fluorescence imaging played complementary roles in this study. PANORAMA provided a label-free approach for overall sEV detection and characterization, allowing for the analysis of sEV population and size distribution. On the other hand, fluorescence imaging with specific markers like *miR-21* enabled a more targeted analysis, focusing on a specific cancer-related biomarker within the sEVs. Together, PANORAMA and fluorescence imaging provided a comprehensive approach for sEV analysis, enabling the ability to explore the potential of sEVs as cancer biomarkers for screening and diagnostic applications.

## Summary and conclusion

Using PANORAMA in combination with fluorescence imaging, we demonstrate the capability of quantifying, sizing, and detecting specific molecular contents, such as *miR-21* within each sEV. This integrated approach not only facilitates the identification of sEV sub-populations but also enables the pinpointing of those originating from cancer cells. This system is able to enumerate with high sensitivity over a broad dynamic range. We capture and retain sEV's using exosome-specific antibody cocktail against sEV surface proteins, providing specificity in the type of vesicles being analyzed by PANORAMA. We demonstrate that the functionalized AGNIS chip retains dye-labeled sEV purified from cell culture supernatant as well as genetically encoded *GFP*-labeled sEVs secreted from lung cancer xenografts in mouse plasma. We find that sEVs released into circulation can be detected from very small tumors. However, the concentration of sEVs does not increase proportionally to tumor size without limit, but rather rapidly reaches steady-state levels even as the tumor enlarges to the point when the animals must be sacrificed, suggesting an elimination or excretion metabolic mechanism that keeps up with the increasing sEV production by growing tumors. We also observe plateauing steady-state levels in cancer patients, and much like what is seen from the preclinical models – the levels of sEVs appear to be at steady-state levels that are ~8-fold higher than the non-cancer state, exhibiting a variable proportion of *miR-21* labeling across different cancer types with 100% cancer specificity. Notably, this elevated level of sEVs shows little variation between individuals or among tumors of different origins or stages, underscoring the robustness and generalizability of our findings. While the physiologic processes responsible for generating this homeostatic status is unclear at this point, the higher steady-state level of sEVs in the blood of individuals could be applicable as a universal biomarker for the presence of cancer. We believe technologies like PANORAMA, which is capable of detecting, quantifying, and characterizing sEVs at high sensitivity could be useful for early cancer detection, even at the earliest stage of cancer development.

## Data availability

All source data supporting this study's findings are available in Supplementary Data 1 & 2, PANORAMA images of the different subgroups is provided in the following GitHub Link: https://github.com/nanobiophotonicsgroup/Data-for-Communications-Medicine.git All other data is available from the corresponding author on reasonable request.

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

## Acknowledgements
We thank Ibrahim Misbah for fabricating the AGNIS substrates used in this research and Heather Marie Otte for helping with the writing.

## Author contributions
W.-C.S. and S.H.L. conceived the idea and directed the study. N.O. and W.-C.S designed the experiment and analyzed the data. N.O. and M.S.M. performed the experiments and produced the figures. J.H. and Y.Q. performed exosome purification and prepared plasma samples. S.F.S., E.H.S., C.T., W.L.H., A.G., M.M.H., K.K.H., S.H.L. provided the plasma samples. M.S.M. and N.L. carried out the study to compare qPCR and our technique. N.O., M.S.M, S.H.L., and W.-C.S. provided substantial input to the project and to the writing of the manuscript.

## Competing interests
S.H.L and W.-C.S. have significant financial interests in Seek Diagnostics Inc. All other authors have no competing interests to declare.
