## [Peer Review File · Communications Medicine]

Reviewers' comments:

Reviewer #1 (Remarks to the Author):

Studies of extracellular vesicles (EV) have revealed that they are heterogeneous in size and molecular composition, which calls for the need to analyze single EVs to unravel their heterogeneity. The authors applied PANORAMA coupled with fluorescence microscopy to analyze individual exosomes in this work. The PANORAMA has been previously published in Nat. Comm (11, 5805, 2020) in their lab. With this technique, they were able to analyze the single EVs, both for determining the size and the number. Using probes to hybridize and label microRNAs contained within exosomes, they were able to simultaneously interrogate the molecular information inside these small vesicles. This enables them to use this technique to analyze cancer-specific exosomes with improved sensitivity and specificity for cancer early diagnostics. I believe the development of single vesicle techniques can reveal unique molecular profiles of cell-specific EVs and will facilitate innovative clinical applications of these vesicles. My suggestions for this work are the major revision and I have below questions:

1. From the abstract, one may feel this work was an application of their previously developed technique so-called PANORAMA. The previous PANORAMA is based on dark field detection and the current one is with fluorescence detection. I would suggest the authors show the novelty and the new features of the current technique in the abstract.
2. The particle size data obtained by Nanosight and PANORAMA (Fig.1e) may not be comparable. The particle measured by Nanosight is the whole vesicle population isolated by UC, while the PANORAMA measured the vesicle subpopulation selected by antibody (CD63). Thus, the results may not support their conclusion described in the text: "Figure 1e shows the size distribution characterized by Nanosight and PANORAMA, where Nanosight reports an average size of 135 ± 102 nm with the smallest particle size at 48 nm while PANORAMA provides an average size of 127 ± 76 nm with the smallest being 34 nm. The majority of the exosomes are well within the 50 to 200 nm range using both sizing methods. However, PANORAMA shows superior sensitivity and sizing resolution for exosomes smaller than 75 nm compared to Nanosight."
3. In Fig. 1, the authors gave two types of scale bars (Scale bar: a, 2 μ m, b, 10 μ m). What does "a" or "b" stand for?
4. On the single exosome detection level, it is highly recommended that the authors compare their method to the current techniques, such as Exoview and Nanoflow. While on cancer detection level, the authors should compare their results to the current techniques regarding sensitivity and specificity, and describe the advantages of using exosomal MIR-21 as the cancer marker instead of plasma-MIR-21 or other detection methods.
5. Have the authors tested other biofluids (such as urine, saliva, etc)? Using the samples with a relatively simple matrix may reduce non-specific bindings and provide stable results.
6. The authors claimed that "...a variable proportion of MIR-21 labeling among different tumor types with 100% cancer specificity." The limitations of this method should be discussed (such as reproducibility). Also, the readers may want to know the potential clinical translations of this method for cancer diagnostics.

Reviewer #2 (Remarks to the Author):

In this study, Ohannesian, et al. sought to design a single exosome characterisation platform for cancer detection. The assay relies capture of exosomes using tetraspannin antibodies and detection using plasmonic nanoaperture label-free imaging (PANORAMA) and fluorescent microscopy. While in principle, single exosome detection and characterisation is challenging and could lead to high resolution diagnostic assays, there are some major concerns and detailed explanations that need to be addressed, and as a result I cannot recommend for publication.

The term “exosome” is not appropriate and implies a particular pathway of EV biogenesis that may not be restricted to EVs of a given size. As per the MISEV 2018 guidelines, the authors should omit the word “exosome”, instead using “small extracellular vesicle (sEV)”.

I am not sure why the authors have used “MIR” to designate miRNA, and not followed the mir/miR/MIR precursor/mature/gene convention for human miRNA.

There is no comparison to commonly used techniques, and is therefore not possible to draw conclusions on the performance of the authors’ strategy. For example, there is no confirmation that hsa-miR-21 is present in sEVs from any clinical or preclinical sample used. This is a critical control that is lacking and should be addressed with an orthogonal technique such as qPCR or ddPCR.

There is a lack of EV characterisation, relying solely on nanosight size characterisation. This does not follow the standard recommendations of the MISEV 2018 guidelines for EV characterisation. The authors should be including EM images of EVs, including EVs captured on the device, and showing that EVs captured do express the expected markers that they are using.

The authors state in line 262 that “cancer detection sensitivity of 99.5% and specificity of 97.3%” but do not provide any ROC curves of their analysis. Given that the authors find that EV concentration alone provides such a significant separation between healthy and cancer patients, what is the use of using the molecular beacon given the poor performance of the beacon to separate healthy and cancer. Furthermore, how do their findings that EV concentration alone stratifies patients fit in with current literature?

The authors do not describe their statistical approaches anywhere in the manuscript.

Reviewer #3 (Remarks to the Author):

In this work, Ohannesian et al. develop an optical analytical technique that can detect and digitally count individual exosomes directly from low-volume human plasma samples. The team tests their method both on samples from cell secretions, cancer cell implanted animals, and healthy and cancer-diagnosed humans. Moreover, by coupling the PANORAMA exosome-detecting platform with the fluorescent MIR-21 detection method, the authors were able to detect cancer before the tumors were large and were able to study and comment on time-resolved exosome level change with tumor growth.

This is an exemplary work on how novel nanophotonic technologies can make an impact in medical diagnostics, specifically on early cancer detection. The experiments are well thought out and the method is tested over an extraordinary amount of positive and negative samples, all delivering consistent results. The method's repeatability is tested extensively and consistent results show strong implications for the transfer of the technology for future clinical use. A large amount of data is presented applying appropriate statistical methods to derive results in this manuscript. The plots in the figures are self-descriptive and clear.

I have one major comment that needs to be clarified before the publication of this manuscript. If this method requires labeling the exosomes to detect MIR-21 for specific tumor-associated exosome detection, then wouldn't the same fluorescent detection data be sufficient to detect exosomes? The authors need to clarify the individual roles of the label-free PANORAMA and labeled fluorescent micro-RNA detection techniques in their work. They are two different detection mechanisms and require different optical instrumentation. Are these methods complementary? How are these two techniques strictly necessary for this method's success? Specifically, in the conclusion, the authors state that it is the PANORAMA that detect, size and molecularly characterize intravesicular MIR content. How is the label-free PANORAMA technique capable of detecting intravascular molecular content? I suggest that the authors add a discussion section to discuss these aspects as well as the major findings of their extensive experimental findings.

Reviewer #4 (Remarks to the Author):

The research lacks novelty as similar work has been conducted before.
<https://doi.org/10.1038/s41467-020-19678-w>

Dear Editor,

We are grateful for handling the review process of our manuscript. Kindly find below the reviewers' comments (written in black) for which we have provided answers and explanation/clarification (written in red).

Sincerely,
Wei-Chuan Shih

Reviewers' comments:

Reviewer #1 (Remarks to the Author):

Studies of extracellular vesicles (EV) have revealed that they are heterogeneous in size and molecular composition, which calls for the need to analyze single EVs to unravel their heterogeneity. The authors applied PANORAMA coupled with fluorescence microscopy to analyze individual sEVs in this work. The PANORAMA has been previously published in Nat. Comm (11, 5805, 2020) in their lab. With this technique, they were able to analyze the single EVs, both for determining the size and the number. Using probes to hybridize and label microRNAs contained within sEVs, they were able to simultaneously interrogate the molecular information inside these small vesicles. This enables them to use this technique to analyze cancer-specific sEVs with improved sensitivity and specificity for cancer early diagnostics. I believe the development of single vesicle techniques can reveal unique molecular profiles of cell specific EVs and will facilitate innovative clinical applications of these vesicles. My suggestions for this work are the major revision and I have below questions:

1. From the abstract, one may feel this work was an application of their previously developed technique so-called PANORAMA. The previous PANORAMA is based on dark field detection and the current one is with fluorescence detection. I would suggest the authors show the novelty and the new features of the current technique in the abstract.

Response: The current study is new in two aspects: 1. It is the first time PANORAMA has been combined with fluorescence imaging as one integral imaging technique; 2. It is the first time the PANORAMA-fluorescence technique has been employed to analyze small extracellular vesicles (sEVs) with significant insight gained. The co-acquisition of PANORAMA and fluorescence images enables the label-free visualization, enumeration, and size determination, as well as the identification of molecular specificity via the fluorescence label. In human blood plasma samples, a numerical increase of sEV regardless of cancer type has been revealed by PANORAMA-derived exosome counts and size distribution. The fluorescence detection of cargo miR-21 provides molecular specificity within the same sEV population at the single unit level, which pinpoints the sEV subset of cancer origin. It's crucial to acknowledge that the elevated sEV counts observed might not solely due to cancer. Conditions beyond malignancy, such as inflammatory or chronic diseases, could potentially lead to similar elevations. While the current study doesn't delve into the specifics of these conditions, it recognizes the significance of this phenomenon as an avenue for future research.

To highlight the novelty of our work in the abstract, we have included the following section P1L24: "In this study, we demonstrate the novel combination of PANORAMA and fluorescence imaging for single sEV analysis for the first time. The co-acquisition of PANORAMA and fluorescence images enables the label-free visualization, enumeration, and size determination, as well as the identification of molecular specificity through fluorescence labeling. PANORAMA-derived sEV counts have shown a numerical increase in sEVs present in cancer patient human blood plasma samples, regardless of cancer type. The fluorescence detection of cargo miR-21 provides molecular specificity within the same sEV population at the single unit level, which pinpoints the sEVs subset of cancer origin."

And to P2L13: "Our techniques and findings could have significant impacts on the understanding of cancer biology as well as the development of new cancer detection and diagnostic technologies."

2. The particle size data obtained by Nanosight and PANORAMA (Fig. 1e) may not be comparable. The particle measured by Nanosight is the whole vesicle population isolated by UC, while the PANORAMA measured the vesicle subpopulation selected by antibody (CD63). Thus, the results may not support their conclusion described in the text: "Figure 1e shows the size distribution characterized by Nanosight and PANORAMA, where Nanosight reports an average size of 135 ± 102 nm with the smallest particle size at 48 nm while PANORAMA provides an average size of 127 ± 76 nm with the smallest being 34 nm. The majority of the sEVs are well within the 50 to 200 nm range using both sizing methods. However, PANORAMA shows superior sensitivity and sizing resolution for sEVs smaller than 75 nm compared to Nanosight."

Response: We acknowledge the distinction between the reported values of Nanosight, which represent the entire vesicle population isolated via ultracentrifugation (UC), and the PANORAMA results, which indicate the size distribution after washing. To emphasize the consistency and agreement between the size distributions obtained from Nanosight and PANORAMA, thereby indicating the reliability of the calibration curve for interpreting vesicle contrast distribution in PANORAMA experiments, we have included the following information in the manuscript on P6L21: "Furthermore, to capture the size distribution of whole vesicle population isolated by UC, we utilized PANORAMA to detect the vesicles on an IgG functionalized surface before washing. We applied the same calibration curve used for Nanosight measurements to determine the vesicle size based on PANORAMA contrast (**Extended Data S1a & S1b**). The average PANORAMA contrast before washing was measured to be $12.54 \pm 5.3\%$, corresponding to a size of 137 ± 68.31 nm (size distribution shown in **Extended Data S1c**). The size distribution of vesicles detected via Nanosight (135 ± 102 nm) and PANORAMA on the IgG functionalized surface (137 ± 68.31 nm) represents the same whole vesicle population obtained through UC. Notably, the size distributions obtained from both techniques exhibit excellent agreement, demonstrating the robustness of the calibration curve. This indicates that the calibration curve can be reliably applied to interpret the contrast distribution of sEV across different experiments conducted using PANORAMA."

In addition to the added text, we added Extended data Figure S1c to the supplementary document to present the calculated size distribution of vesicles detected via PANORAMA on an IgG functionalized surface before washing.

3. In Fig. 1, the authors gave two types of scale bars (Scale bar: a, 2 μm , b, 10 μm). What does “a” or “b” stand for?

Response: Fig. 1a illustrates the time-evolution of sEV detection using PANORAMA, showcasing both selected region and full field of view images. The first set of images in the selected region highlights specific areas of interest and includes a scale bar measuring 2 μm for reference. The last two images in Fig. 1a provide the full field of view and are accompanied by a scale bar measuring 10 μm . Fig. 1b presents the quantification of sEVs detected over time, focusing on the selected section from the full field of view reported in Fig. 1a.

For clarification we have modified the description of the Figure 1 on P26L3 as follows: “Scale bar: **a**, 2 μm for selected region images; 10 μm for full field of view images.”

4. On the single sEV detection level, it is highly recommended that the authors compare their method to the current techniques, such as Exoview and Nanoflow. While on cancer detection level, the authors should compare their results to the current techniques regarding sensitivity and specificity and describe the advantages of using exosomal MIR-21 as the cancer marker instead of plasma-MIR-21 or other detection methods.

Response: Since we have no access to these instruments, a direct comparison with these specific tools is currently not feasible and is outside the scope of our study. We aim to report a new method using PANORAMA which offers valuable insights and contributes to the existing knowledge of single sEV detection. We recognize the importance of future research and collaborations with groups having access to Exoview and Nanoflow, as it could enable further comparisons and validation of our method.

It is well known that sEVs contain high levels of miR, and the miR content in the sEVs reflect the cell of origin, whether it is from normal organs or from tumor cells. Free miR not encapsulated in sEVs certainly will be present that are shed from dying or lysed cells, but that would be degraded by nucleases rather than being protected in sEVs. MiRs inside sEVs are much more stable and less subject to degradation, and therefore could be analyzed very easily using molecular beacon hybridization. Since we are visualizing each sEV, we should also be able to characterize the miR content in each of the sEVs, along with proteins. A key advantage is that we will be able to quantify the proportion of sEVs that are cancer derived vs. normal cell derived based on assessing the miR in each of the sEVs that are immobilized on the AGNIS chip. We have added this section on page P7L14” It is important to note that utilizing plasma miR-21 is conceptually similar to detecting CTDNA, which can offer high specificity and sensitivity for cancer detection. However, there are notable limitations when using CTDNA for cancer screening. The lack of early-stage DNA shedding poses challenges for early diagnosis. The low abundance of CTDNA in the bloodstream and its susceptibility to degradation further complicate reliable detection and quantification. Variability in CTDNA levels over time and their dependency on factors like tumor size and treatment introduce difficulties in obtaining consistent results. The cost and technical requirements of advanced technologies like next-generation sequencing can limit the accessibility of CTDNA analysis. Furthermore, plasma miRs are prone to be degraded by nucleases rather than being protected in sEVs. In contrast, sEVs offer advantages in their

abundance and capacity of storing genetic material for extended periods, making them good candidates for biomarker analysis. Therefore, utilizing sEV miR-21 could potentially provide enhanced sensitivity and reproducibility to detect cancer at early stages, all at a significantly lower cost.”

5. Have the authors tested other biofluids (such as urine, saliva, etc)? Using the samples with a relatively simple matrix may reduce non-specific bindings and provide stable results.

Response: This is outside the scope of this study since we have neither collected those samples nor have the availability of those samples that match the blood samples collected from patients. Our investigation focuses primarily on evaluating sEVs in blood/plasma as an early diagnostic blood test to differentiate healthy individuals from those with cancer. While the potential application of the platform to other biofluids is acknowledged, the paper's objective was not to compare the detectability of sEVs across various biofluids.

Notably, we examined serum, heparinized plasma, and EDTA plasma samples collected simultaneously from the same patients. We observed that heparinized and EDTA plasma exhibited similar sEV levels while serum levels were slightly lower. This discrepancy may be attributed to the clotting that occurs in the red top tubes during serum collection, potentially leading to the entrapment of sEVs and subsequently reducing their concentration compared to the original sample.

6. The authors claimed that “...a variable proportion of MIR-21 labeling among different tumor types with 100% cancer specificity.” The limitations of this method should be discussed (such as reproducibility). Also, the readers may want to know the potential clinical translations of this method for cancer diagnostics.

Response: Reviewer 3 also raises a similar concern which we have included a discussion section in the study to address the limitations of sEV miR-21 as a cancer detection technique. This section is now included in the manuscript on P14L10.

Reviewer #2 (Remarks to the Author):

In this study, Ohannesian, et al. sought to design a single sEV characterization platform for cancer detection. The assay relies on capture of sEVs using tetraspannin antibodies and detection using plasmonic nanoaperture label-free imaging (PANORAMA) and fluorescent microscopy. While in principle, single sEV detection and characterization is challenging and could lead to high resolution diagnostic assays, there are some major concerns and detailed explanations that need to be addressed, and as a result I cannot recommend for publication.

The term “sEV” is not appropriate and implies a particular pathway of EV biogenesis that may not be restricted to EVs of a given size. As per the MIsEV 2018 guidelines, the authors should omit the word “sEV”, instead using “small extracellular vesicle (sEV)”.

Response: We appreciate the reviewer's concern regarding the use of terminology. We understand the rationale behind the MIsEV 2018 guidelines, which aim to promote consistency and precision in EV research. We have thus taken the necessary steps to address this concern. The term "exosome" has been substituted with "small extracellular vesicle (sEV)", aligning with the guidelines' recommendations. This modification is in accordance with the goal of avoiding implications of a specific biogenesis pathway for EVs of a particular size, as highlighted by the MIsEV 2018 guidelines.

I am not sure why the authors have used "MIR" to designate miRNA, and not followed the mir/miR/MIR precursor/mature/gene convention for human miRNA.

Response: We have changed the nomenclature in the paper to reflect the reviewer's recommendation.

There is no comparison to commonly used techniques and is therefore not possible to draw conclusions on the performance of the authors' strategy. For example, there is no confirmation that hsa-miR-21 is present in sEVs from any clinical or preclinical sample used. This is a critical control that is lacking and should be addressed with an orthogonal technique such as qPCR or ddPCR.

Response: We used molecular beacon probes that targets specific miRNA targets since it is the only type of molecular detection that could resolve single sEV label, not possible with QPCR or ELISA tests. Those are bulk methods and do not help validate our platform. We have used cell lines known to contain or lack miR-21 expression based on numerous studies, and validated using molecular beacon which is a well-established method. The goal of our study is not to compare QPCR with molecular beacon detection since the latter method is highly specific and allows individual sEV visualization which is what we need for our assay. After showing that our technique was specific, we were able to show using both in vitro cell line sEVs and also preclinical mouse tumor xenograft of lung cancer cell lines that the miR-21 is extremely specific. At the level of 10-20 microliters of plasma, there are no QPCR methods sensitive enough to detect the microRNA levels at such small volumes without completely bleeding and sacrificing the animals at specific longitudinal timepoints. Using our method, we obtain a small drop of blood from the facial vein of each animal, captured the blood in EDTA tubes, centrifuged to collect the plasma, and kept the plasma in the freezer until all the samples over the course of several weeks have been collected for the one-time analysis.

There is a lack of EV characterisation, relying solely on nanosight size characterisation. This does not follow the standard recommendations of the MIsEV 2018 guidelines for EV characterisation. The authors should be including EM images of EVs, including EVs captured on the device, and showing that EVs captured do express the expected markers that they are using.

Response: We understand the reviewer's concern regarding the lack of sEV characterization in our study. Currently, we are working on performing SEM profiling of the sEVs, but our current ability to carry out this procedure is hindered by the unavailability of the necessary fixation chemicals. However, we employed a widely used extracellular vesicle (EV) extraction method, specifically ultracentrifugation, to isolate the desired sEVs. sEVs of interest are exosomes that

range in size from approximately 30 to 200 nm. To validate the presence of sEVs within our samples and confirm their size range, we conducted Nanosight analysis. Furthermore, we utilized sEV capture via tetraspanin proteins, which are highly expressed on exosomes as documented in the literature that are responsible for intercellular communication. We selected antibodies that target prominent tetraspanins found on sEVs, irrespective of their origin. References 16 to 18 support our rationale for choosing these antibodies, providing examples of tetraspanin expression on sEVs. We have also added the following sentence on P3L6 for more clarity:” Exosomes are sEVs ranging from 30 to 200 nm responsible for intercellular communication.”

The authors state in line 262 that “cancer detection sensitivity of 99.5% and specificity of 97.3%” but do not provide any ROC curves of their analysis. Given that the authors find that EV concentration alone provides such a significant separation between healthy and cancer patients, what is the use of using the molecular beacon given the poor performance of the beacon to separate healthy and cancer. Furthermore, how do their findings that EV concentration alone stratifies patients fit in with current literature?

Response: As suggested by the reviewer, we have added ROC analysis in the supplemental materials. While EV concentration appears to be able to stratify the absence and presence of cancer in patients, adding biomarkers such as miRNA detection provides the 1) validation needed that any elevation that occur are indeed from the cancer cells that are actively secreting the sEVs, 2) molecular characterization of sEVs is feasible using molecular beacon detection on our platform to couple PANORAMA with fluorescence imaging, and 3) molecular characterization of the sEVs will provide diagnostic accuracy in states where sEV count elevation is seen that is not cancer-specific. To our knowledge, we believe this is the first study that has shown that cancer patients can be stratified from healthy individuals from simply enumerating the concentration of sEVs in a simple drop of blood. We have proven this using preclinical models that identify very low background levels of circulating sEVs that are CD9/CD63/CD81+ and negative for miR-21 in non-tumor bearing mice, but as soon as there is any palpable evidence of tumor, there is concomitant increase in the specific sEV levels that is quantifiable based on PANORAMA but also with biomarker verification. Using longitudinal sampling with weekly blood drawn from the mice, we can see this elevation increase and plateau which indicates a steady state that is achieved in the animals. The steady state can also be seen in the patients, since regardless of stage or tumor type, the mean levels of sEVs appear to be similar, but at a level that is nearly 10-fold higher than healthy individuals. These are novel observations not previously reported. There are reports using ELISA similar observations (Ref. 20), but not the quantitative enumeration that we have reported given the techniques used by the other studies.

The authors do not describe their statistical approaches anywhere in the manuscript.

Response: The details of the statistical methods used are now added to the methods section (P21L23).

Reviewer #3 (Remarks to the Author):

In this work, Ohannesian et al. develop an optical analytical technique that can detect and

digitally count individual sEVs directly from low-volume human plasma samples. The team tests their method both on samples from cell secretions, cancer cell implanted animals, and healthy and cancer-diagnosed humans. Moreover, by coupling the PANORAMA sEV-detecting platform with the fluorescent miR-21 detection method, the authors were able to detect cancer before the tumors were large and were able to study and comment on time-resolved sEV level change with tumor growth.

This is an exemplary work on how novel nanophotonic technologies can make an impact in medical diagnostics, specifically on early cancer detection. The experiments are well thought out and the method is tested over an extraordinary amount of positive and negative samples, all delivering consistent results. The method's repeatability is tested extensively, and consistent results show strong implications for the transfer of the technology for future clinical use. A large amount of data is presented applying appropriate statistical methods to derive results in this manuscript. The plots in the figures are self-descriptive and clear.

I have one major comment that needs to be clarified before the publication of this manuscript. If this method requires labeling the sEVs to detect miR-21 for specific tumor-associated sEV detection, then wouldn't the same fluorescent detection data be sufficient to detect sEVs? The authors need to clarify the individual roles of the label-free PANORAMA and labeled fluorescent micro-RNA detection techniques in their work. They are two different detection mechanisms and require different optical instrumentation. Are these methods complementary? How are these two techniques strictly necessary for this method's success? Specifically, in the conclusion, the authors state that it is the PANORAMA that detect, size and molecularly characterize intravesicular miR content. How is the label-free PANORAMA technique capable of detecting intravascular molecular content? I suggest that the authors add a discussion section to discuss these aspects as well as the major findings of their extensive experimental findings.

Response: We appreciate the reviewer's suggestion to include a discussion section and summarize the key findings of the study. While the summary and conclusion sections already provide specific details about the major findings, we have now added a dedicated discussion section. In this section, we will present the roles of PANORAMA and Fluorescence imaging techniques, as well as discuss their limitations on PI3L10 with the following paragraphs: "The role of PANORAMA in this study was to serve as a label-free imaging system for detecting, sizing and quantifying sEVs in human plasma samples. PANORAMA facilitated the visualization and enumeration of sEVs based on their distinct optical properties, eliminating the need for exogenous labels or modifications. By utilizing PANORAMA, we captured real-time images of sEVs and analyzed their characteristics, such as size, providing valuable insights into the sEV population in the samples. Furthermore, the distinct sEV levels quantified by PANORAMA from diseased and healthy populations have been clearly observed in animal models and human subjects. However, it's important to note that sEVs count may also be elevated with non-cancer conditions, such as inflammatory conditions or chronic diseases, an area ripe for future research. Therefore, the PANORAMA established protocol could be used as a sensitive initial screening technique to identify any discrepancies within the patient, further investigations and diagnostic tests are necessary to confirm the nature of cancer using specific biomarkers that can distinguish cancerous vs non-cancerous states.

Fluorescence played a crucial role in this study by specifically detecting and quantifying the presence of miR-21, an example of one well-established cancer biomarker, within the identified sEVs. Fluorescence imaging enabled the selective labeling and visualization of miR-21 within the sEV population, providing additional information about the presence and abundance of this specific biomarker. However, it is important to note that not all cancers express the same genetic biomarkers. Moreover, the occurrence of miR-21 in plasma samples from different cancer subsets has shown variations. Thus, relying solely on miR-21 for cancer screening may have limitations and lack the sensitivity to detect all types of cancer, potentially being restricted to a few specific types.

Overall, the combination of PANORAMA and fluorescence imaging played complementary roles in this study. PANORAMA provided a label-free approach for overall sEV detection and characterization, allowing for the analysis of sEV population and size distribution. On the other hand, fluorescence imaging with specific markers like miR-21 enabled a more targeted analysis, focusing on a specific cancer-related biomarker within the sEVs. Together, PANORAMA and fluorescence imaging provided a comprehensive approach for sEV analysis, enabling the ability to explore the potential of sEVs as cancer biomarkers for screening and diagnostic applications.”

Reviewer #4 (Remarks to the Author):

The research lacks novelty as similar work has been conducted before.

<https://doi.org/10.1038/s41467-020-19678-w>

Response: Our previous work was a technical paper showing the capability of PANORAMA to detect particles, which includes extracellular vesicles. There was little to no applications in the initial paper. The current paper is an application of the capability of the technique for application in small extracellular vesicle (sEV) detection, including technical optimization, preclinical and clinical validation of the ability of surmise cancer state with sEV enumeration in a small volume of plasma, highlighting that the concentration of circulating CD9/CD81/CD63+ EVs are substantially higher in cancer patients that we were able to show in preclinical models that reflect what we see in patients. This novelty has never been demonstrated previously in other publications, demonstrating the uniqueness of this work not previously shown in our work or others.

Reviewers' comments:

Reviewer #1 (Remarks to the Author):

The authors have revised manuscript and addressed all my comments. Now it can be published either as is or subject to minor revisions as indicated. In abstract, line 24, please avoid using the word “novel” through out the manuscript.

Reviewer #2 (Remarks to the Author):

The authors have addressed most of my concerns. However, the authors have not addressed the critical control of demonstrating the miR-21 is present in their sEVs using an orthogonal technique such as qPCR. The authors are correct that their technology allows for the resolution of single sEV counting, thereby providing a clear advantage over a bulk measurement. However, the concern is not about measuring sEV heterogeneity, but rather a standard confirmation that this particular miRNA is present in their samples so that the specificity of their assay is confirmed. This can easily be carried out using in vitro-derived sEVs and qPCR. The fact that unreferenced studies confirm miR-21 is present in cells, is not related to miR-21 being present in their sEVs that they have isolated. This wouldn't be required to do in clinical or mouse work.

Furthermore, the fact the authors are claiming that qPCR is not sensitive enough to measure miRNA in their animal work, yet their system can measure an occurrence of positive sEVs up to 40% would imply their technology is far more sensitive than qPCR – an impressive claim that should be validated.

Reviewer #3 (Remarks to the Author):

The authors responded to the reviewer's comments in detail, which clarified the significant confusion on the novelty of this work wrt their previous technology-oriented paper and with alternative technologies available in the market. I recommend the publication of this manuscript.

We thank the reviewers for their comments, and we address them point by point below.

Reviewers' comments:

Reviewer #1 (Remarks to the Author):

The authors have revised manuscript and addressed all my comments. Now it can be published either as is or subject to minor revisions as indicated. In abstract, line 24, please avoid using the word "novel" through out the manuscript.

Response: Removed.

Reviewer #2 (Remarks to the Author):

The authors have addressed most of my concerns. However, the authors have not addressed the critical control of demonstrating the miR-21 is present in their sEVs using an orthogonal technique such as qPCR. The authors are correct that their technology allows for the resolution of single sEV counting, thereby providing a clear advantage over a bulk measurement. However, the concern is not about measuring sEV heterogeneity, but rather a standard confirmation that this particular miRNA is present in their samples so that the specificity of their assay is confirmed. This can easily be carried out using in vitro-derived sEVs and qPCR. The fact that unreferenced studies confirm miR-21 is present in cells, is not related to miR-21 being present in their sEVs that they have isolated. This wouldn't be required to do in clinical or mouse work.

Furthermore, the fact the authors are claiming that qPCR is not sensitive enough to measure miRNA in their animal work, yet their system can measure an occurrence of positive sEVs up to 40% would imply their technology is far more sensitive than qPCR – an impressive claim that should be validated.

Response:

We have conducted an additional study to address reviewer 2's concerns. We have performed miR-21 detection/quantitation using RT-qPCR vs. our technique and the results are included in the revised manuscript (Page 9, Line 8; Page 19, Line 4 and Line 13; Figure 2(c)(d)). They are reproduced on the next page for your reference.

To briefly summarize, we have confirmed that the miR-21 are present and can be quantified in cell culture-derived sEVs using RT-qPCR (addressing concern #1). Furthermore, we have compared the limit of detection between RT-qPCR and our technique and found that our technique provides about 5,000-50,000 fold better limit of detection (10^7 sEV count for qPCR vs. 2×10^2 (H460) and 2×10^3 (293A) sEV count for our technique) (addressing concern #2). We believe both these findings directly address the reviewer's two concerns, and we would like to thank him/her for the suggestions because these results would definitely strengthen the paper.

Page 9, Line 8

..... We further conducted a comprehensive comparison of our findings using quantitative polymerase chain reaction (qPCR) and PANORAMA, where experiments involved varying concentrations of sEVs derived from H460 and 293A cell lines. The qPCR results indicated a 7.6-fold difference in C_T values between H460 and 293A sEVs at the highest concentration tested (1×10^8 particles/reaction). The lowest detectable occurrence of miR-21 was observed at a concentration of 1×10^7 particles/reaction from both cell lines (**Fig. 2c**). On the other hand,

PANORAMA experiments were conducted using 20 μL of sEV solutions at different concentrations, mirroring the concentrations used in qPCR. **Figure 2d** shows that the lowest detectable miR-21 occurrence for H460 was at a concentration of 1×10^4 particles/mL (equivalent to 200 particles/experiment), while for 293A, it was at a concentration of 1×10^5 particles/mL (equivalent to 2000 particles/experiment). The miR-21 occurrence for H460 varied from 50-79% for all different concentrations, and for 293A, it varied from 6-16% across all concentrations. Notably, PANORAMA demonstrated the capability to detect and quantify sEVs at a concentration as low as 1×10^4 particles/mL from both cell lines, corresponding to a limit of detection (LOD) of 16.7 attom for sEVs (**Fig. 2d**). These results highlight the enhanced sensitivity of PANORAMA in comparison to qPCR, particularly in the detection of miR-21 occurrence at lower concentrations. Furthermore, the exact subset of miR-21-positive sEVs can be examined at the single unit level by the combination of PANORAMA and fluorescence imaging.

Page 19, Line 4

sEV isolation from cell culture medium for qPCR

H460 and 293A cells were cultured in RPMI1640/DMEM with 10% FBS, then the serum-containing cell culture medium was removed, and the cells were cultured in an exosome free culture medium for 48 h. Then, the cell-culture supernatants were centrifuged at $2000 \times g$ for 10 min to remove debris, and large vesicles and apoptotic bodies were removed by centrifugation at $10,000 \times g$ for 30 min. Next, the sEVs were enriched by centrifugation at $3,000 \times g$ for 30 min with the Amicon[®] Ultra Centrifugal Filter (Sigma catalog UFC9100) and then purified using Total Exosome Isolation Reagent (Thermo Fisher catalog 4478359) according to the manufacturer's recommendation. Isolated sEVs were resuspended with cold PBS and analyzed using a NanoSight NS300 device (Malvern, PA, USA).

Page 19, Line 13

RNA isolation from sEVs and RT-qPCR

Purified sEVs from H460 and 293A cells were subjected to Total Exosome RNA & Protein Isolation Kit (Thermo Fisher catalog 4478545) to extract total RNA. Followed by RT-PCR using OneTaq[®] RT-PCR Kit (New England Biolabs catalog E5310S).

Real-time qPCR was done with the PowerUp[™] SYBR[™] Green Master Mix (Thermo Fisher catalog A25741). The thermal cycling protocol was as follows: initial denaturation for 10 min at 95°C , followed by 50 cycles of 15 s at 95°C and 60 s at 60°C . The primer sequences are listed as follows:

miR-21 for RT: 5'-CTCAACTGGTGTCTGGAGTCGGCAATTCAGTTGAGTCAACATC-3'

miR-21 for qPCR: 5'- ACACTCCAGCTGGGTAGCTTATCAGACTGA-3'

Figure 2(c)(d))

..... **c.** miR-21 occurrence at various concentrations of H460 & 293A sEVs using qPCR. **d.** miR-21 occurrence and detected sEV count at various concentrations of H460 & 293A sEVs using PANORAMA. Scale bar: 10 μ m.

Reviewer #3 (Remarks to the Author):

The authors responded to the reviewer's comments in detail, which clarified the significant confusion on the novelty of this work wrt their previous technology-oriented paper and with alternative technologies available in the market. I recommend the publication of this manuscript.

Yours Sincerely,

Wei-Chuan Shih, Ph.D.

Cullen Engineering Professor

Department of Electrical & Computer Engineering, Biomedical Engineering, Chemistry, and
Materials Science & Engineering

University of Houston

4800 Calhoun Road, Houston TX 77204 USA

TEL: 713-743-4454

E-mail: wshih@uh.edu

REVIEWERS' COMMENTS:

Reviewer #2 (Remarks to the Author):

The authors have addressed all my comments.